# Ameliorative Effect of Ethanolic *Echinacea purpurea* against Hyperthyroidism-Induced Oxidative Stress via AMRK and PPAR Signal Pathway Using Transcriptomics and Network Pharmacology Analysis

**DOI:** 10.3390/ijms24010187

**Published:** 2022-12-22

**Authors:** Yingli Zhu, Jianjun Zhang, Chun Wang, Ting Zheng, Songrui Di, Yinyin Wang, Wenting Fei, Weican Liang, Linyuan Wang

**Affiliations:** 1School of Chinese Materia Medica, Beijing University of Chinese Medicine, Beijing 100029, China; 2School of Traditional Chinese Medicine, Beijing University of Chinese Medicine, Beijing 100029, China; 3School of Traditional Chinese Pharmacy, China Pharmaceutical University, Nanjing 210009, China

**Keywords:** *Echinacea purpurea* (L.) Moench, ethanolic *Echinacea purpurea* extract, thyroxine, oxidative stress, transcriptomic, network pharmacology

## Abstract

*Echinacea purpurea* (L.) Moench (EP) is a well-known botanical supplement with antioxidant characteristics. However, the effects of EP on oxidative stress induced by hyperthyroidism have not yet been studied. This study was designed to evaluate the antioxidative effect of ethanolic *Echinacea Purpurea* (EEP) on hyperthyroidism-induced oxidative stress mice using an integrated strategy combining transcriptomics with network pharmacology analysis. Firstly, a hyperthyroidism mice model was induced via thyroxine (160 mg/kg) and EEP (1, 2, or 4 g/kg) once daily for 2 weeks. Body weight, thyroid-stimulating hormones, and oxidative stress markers were tested. Secondly, EEP regulating the potential genes at transcript level were analyzed. Thirdly, a network pharmacology based on the constituents of EEP identified using UPLC-Q-TOF-MS analysis was adopted. Finally, a joint analysis was performed to identify the key pathway. The results showed that EEP significantly changed the thyroid-stimulating hormones and oxidative stress markers. Meanwhile, RT-qPCR and Western Blotting demonstrated that the mechanism of the antioxidant effect of EEP reversed the mRNA expression of EHHADH, HMGCR and SLC27A2 and the protein expression of FABP and HMGCR in AMPK and PPAR signaling pathways. This study integrates transcriptomics with network pharmacology to reveal the mechanism of ameliorative effect of EEP on hyperthyroidism-induced oxidative stress.

## 1. Introduction

*Echinacea purpurea* (L.) Moench (EP) is a North American medicinal herb [1], with well-documented immune-regulatory, antioxidant, anti-inflammatory, antibacterial, antiviral, antiproliferative, hypoglycemic, antihypertensive, and antiobesity activities [2,3]. According to the literature, EP can be used to treat wounds, burns, bug bites, toothaches, snake bites, skin conditions, and epilepsy. In addition, EP supports the immune system of the body [1]. Importantly, it is also one of the most widely used botanical supplements in the world. The results of a stratified random sample of an adult general health survey showed that 14.7% of 1999 respondents of a large health maintenance organization in northern California frequently used EP [4]. EP was the most frequently used botanical among 739 users of selected plant food supplements in eight different age and sex subgroups in the Netherlands [5]. It should be noted that a high percentage of individuals use EP-containing products, including healthy adults, adults with chronic diseases such as diabetes, and those with metabolic and endocrine function with easily distinguished disorder stages, such as puberty, perimenopause, and pregnancy. Clinical studies show that 8.9% of 20 pregnant women [6], 79% of 1280 adolescents [7], patients with diabetes mellitus [8], and some pre-and postmenopausal women [9] have used EP in their lifetime to alleviate disease symptoms, prevent disease, improve quality of life, and reduce menopausal symptoms. It is noteworthy that endocrine disorders play a vital role in these diseases, which can cause diabetes, thyroid disease, growth disorders, sexual dysfunction, and other hormone-related disorders [10].

Hyperthyroidism is a common endocrine disorder condition (0.78% in China, 0.8% in Europe, and 1.3% in the USA) [11,12,13] and is characterized by an elevated basal metabolic rate caused by an excess of thyroid hormones [14]. Thyroid hormones increase basal metabolic rate and oxidative stress by inducing mitochondrial enzymes [15], and hyperthyroidism increases the formation of reactive oxygen species (ROS) and causes changes in the antioxidant mechanisms of different tissues [16]. Studies have shown that thyroid hormones cause oxidative stress in different tissues [17], and if cellular mechanisms fail to scavenge these reactive oxygen species, toxicity occurs with lipid peroxidation in the liver, heart, muscle, and testes [18]. Numerous studies have demonstrated that melatonin [19], vitamin E [20], Caffeic acid phenylethyl ester [21], Vitamin D [22], etc., can prevent oxidative damage in hyperthyroidism and reduce lipid peroxidation caused by various conditions through their free radical scavenging effects and by directly increasing the activity of antioxidants. However, to date, studies on the effects of EP, a well-known potent antioxidant, on oxidative stress induced by hyperthyroidism have not yet been studied.

The important components of EP are alkylamides, polysaccharides, glycoproteins, flavonoids, and phenolic compounds. Phenolic compounds include derivatives of caffeic acid, such as chicoric acid, caftaric acid, and coutaric acid [23]. EP supplements are generally sold as encapsulated tablets containing aerial parts or dried roots or as tablets containing extruded material from pressed plants or ethanol extracts [1]. Total phenols, caffeic acid derivative contents, phenolic compounds (such as chicoric acid and caffeic acid), and flavonoids (such as nicotiflorin and rutin) are present in ethanol extracts [24,25], and the total phenols and caffeic acid derivatives in the ethanolic extract of *Echinacea purpurea* have been shown to have antioxidant effects in vitro [26]. However, to our knowledge, there have been no studies on the antioxidative effects of hyperthyroidism-induced oxidative stress of the ethanolic EP (EEP) extract in terms of evaluating the total content of phenols, caffeic acid derivatives, and flavonoids. Thus, we propose that EEP may have antioxidative effects against oxidative stress caused by hyperthyroidism in mice.

Network pharmacology allows the intervention and influence of the chemical compounds of EEP on the oxidative stress network induced by hyperthyroidism to be studied from multiple perspectives at the molecular level by constructing a multicompound, multitarget, and multipathway interaction network. However, network pharmacology is based on public databases and can only predict potential outcomes [27]. Transcriptomic sequencing analysis allows for the identification and visualization of changes in gene expression profiles during the development of drug-interfered disease [28], and high-throughput sequencing-based transcriptomic analysis provides a way to rapidly identify the full landscape of mRNA changes [29] in EEP resistance to hyperthyroidism-induced oxidative stress and helps to identify mechanisms. Thus, the integration of network pharmacology and transcriptomics can help overcome the limitations of the former lacking an experimental basis and the latter lacking a molecular mechanistic explanation and can contribute to a better understanding of the mechanism of antioxidant effects of EEP against hyperthyroidism-induced oxidative stress.

In this study, an integrated strategy combining network pharmacology and transcriptomics was used to reveal the antioxidant mechanisms underlying the effects of EEP on hyperthyroidism-induced oxidative stress. The AMPK and PPAR signaling pathways were present in the network pharmacology and transcriptomics results. Recent studies have reported that activation of the AMPK pathway enhances the activity of antioxidant enzymes to attenuate oxidative stress [30,31], and the PPAR pathway also has an important role in the regulation of oxidative stress processes [32]. Studies have shown that increased ROS production induces mitochondrial dysfunction and plays a key role in oxidative stress [33]. AMPK is involved in intracellular glucose and lipid metabolism, and studies have shown that it can be interfered with by antioxidants and activated by ROS production [34]. In addition, peroxisome proliferator-activated receptors (PPARs) are nuclear receptors that act as ligand-dependent transcription factors and can be activated by fatty acids (FAs). Three isoforms of PPARs have been identified (namely PPARα, PPARβ/δ and PPARγ), which are capable of binding long-chain polyunsaturated fatty acids (LCPUFAs) [35]. The binding of FAs can trigger the transcription of specific genes including those encoding various metabolic and cellular processes such as beta-oxidation of FAs and lipogenesis and are therefore key mediators of lipid homeostasis [36]. Thus, we suggest that these two pathways may be necessary for EEP to ameliorate hyperthyroidism-induced oxidative stress.

To our knowledge, this is the first study to use an integrated multidatabase network pharmacology approach based on constituents of EEP that were identified via UPLC-Q-TOF-MS analysis combined with high-throughput transcriptomics to explore the mechanism of the ameliorative effect of EEP against oxidative stress caused by hyperthyroidism and provides a deep understanding of the antioxidant mechanism of EEP.

## 2. Results

### 2.1. UPLC-Q-TOF-MS Analysis

In UPLC-Q-TOF-MS analysis of EEP extracts, 18 compounds, including Caftaric acid; Coutaric acid; Cis-fertaric acid; Rhodioloside D; Trans-fertaric acid; N-Acetyl-DL-tryptophan; Chicoric acid; Quercetin 3-O-robinobioside; Rutin; Isoquercitrin; Trihydroxyoctadecanoic acid; 2,4-Undecadiene-8,10-diynoic acid isobutylamide; Dodeca-2E,4E,8Z,10E-tetraenoic acid isobutylamide; Dodeca-2E,4E,8Z,10Z-tetraenoic acid isobutylamide; Hydroxyhexadecanoic acid; Linolenic acid; Linoleic acid; and Palmitic acid were identified through the comparison of retention times of authentic standards. The results are shown in Figure 1 and the full list of compounds are shown in Appendix A.

### 2.2. Effects of EEP on Body Weight Changes and Different Organ Weights in Hyperthyroidism Mice

At the end of the experiment, the body weights of the mice in the model group decreased significantly, compared to those of the control group (*p* < 0.05). Furthermore, the body weights of the mice in the PTU treated group (*p* < 0.05), the EEP-M treated group (*p* < 0.05), and the EEP-H treated group (*p* < 0.05) increased compared to the model group. The spleen weights index in the model group rats decreased significantly compared to the control group (*p* < 0.05), while EEP-M treatment (*p* < 0.01) increased spleen weight index. However, EEP did not change the weight of the thymus glands (Figure 2).

### 2.3. Effects of EEP on Thyroid Hormone Levels in Hyperthyroidism Mice

As shown in Figure 3, TSH levels were significantly decreased in the hyperthyroidism model group compared to the control group (*p* < 0.05), the EEP-M group (*p* < 0.01) and the EEP-H group (*p* < 0.05) were found to significantly increase TSH levels in hyperthyroid mice to levels similar to those of the PTU treated group (*p* < 0.05). T3 levels increased significantly in the hyperthyroid group (*p* < 0.05), while the EEP-M (*p* < 0.01) and the EEP-H (*p* < 0.001) groups showed significantly decreased T3. However, T4 levels did not increase significantly. These results indicate that EEP can improve the symptoms of hyperthyroidism through the regulation of thyroid hormone imbalance.

### 2.4. Effects of EEP on Antioxidative Markers in Hyperthyroidism Mice

As a representative product of lipid peroxidation, the level of MDA reflects the severity of oxidative stress and the severity of free radical attack on cells. Based on the results shown in Figure 4, EEP can inhibit the increase of MDA level in hyperthyroidism-induce mice after administration. MDA levels increased significantly in the hyperthyroid group compared to the normal group (*p* < 0.01), while EEP-L (*p* < 0.05), EEP-M (*p* < 0.05), and EEP-H (*p* < 0.01) significantly decreased MDA levels. On the other hand, antioxidant enzymes can prevent the harmful effects of oxidative stress, such as SOD and CAT, and their oxidative activities represent the level of antioxidant capacity of EEP. SOD levels (*p* < 0.05) and CAT levels (*p* < 0.001) were all significantly decreased in the hyperthyroidism model group compared to the normal group, while EEP-L (*p* < 0.001), EEP-M (*p* < 0.001), and EEP-H (*p* < 0.001) were found to significantly increase SOD levels and CAT levels in hyperthyroid mice.

### 2.5. Transcriptomics Analysis the Antioxidative Mechanism of EEP in Hyperthyroidism Mice

#### 2.5.1. Identification Related Antioxidative Genes of Differentially Expressed Genes of EEP in Hyperthyroidism Mice

Using the Edge R package, 78 DEGs were identified in the hyperthyroidism model versus EEP, 31 of which were upregulated and 47 were downregulated. DEGs and oxidative stress genes were filtered and visualized by the volcano plot shown in Figure 5A. Finally, 43 antioxidative genes caused by hyperthyroidism of EEP have been identified, called intersection genes in Figure 5B.

#### 2.5.2. GO and KEGG Pathway Enrichment Analysis of Related Antioxidative Perturbation Genes of EEP

In the GO functional enrichment analysis, the comparison of the hyperthyroidism model and EEP-treated animals showed that 35 BP terms, 10 CC terms, and 23 MF terms were significantly enriched in antioxidative perturbation genes of EEP. The results of the top 10 GO analysis revealed that in BP, antioxidative perturbation genes of EEP were abundant in the fatty acid metabolic signaling pathway. In CC, antioxidative perturbation genes of EEP were abundant in the peroxisome, mitochondrion, and peroxisomal membrane. In MF, antioxidative perturbation genes of EEP were abundant in myristoyl-CoA hydrolase activity, palmitoyl-CoA hydrolase activity, and acyl-CoA hydrolase activity (Figure 6A). Furthermore, the results of the KEGG analysis showed that the antioxidative perturbation genes of EEP were significantly enriched in the PPAR signaling pathway, fatty acid degradation, fatty acid metabolism, and other pathways (Figure 6B).

#### 2.5.3. Identification and Validation of PPI Network and Related Key Antioxidative Genes of EEP

Antioxidative perturbation genes of EEP were submitted to the STRING database to obtain PPI data and the cytoHubba plugin in Cytoscape 3.9.1 were used to determine associated gene expression in Figure 7A. First, we performed module analysis and identified 20 hub genes as EEP-induced genes via cluster analysis in Figure 7B. The key 10 genes (EHHADH, ACOT2, ACOT8, SLC27A2, EPHX2, HMGCR, ACACA, ACSL4, ACADM and FASN) among the EEP altered genes network were obtained after the analysis of the cluster algorithm and further validated using RT-qPCR (Figure 7C). Finally, EHHADH, ACOT2, ACOT8, SLC27A2, EPHX2, HMGCR and ACACA were confirmed to be expressed in hyperthyroid model mice liver (Figure 7C).

### 2.6. Network Pharmacology Analysis of the Antioxidative Mechanism of EEP 

#### 2.6.1. Prediction of the Target Proteins of EEP and Construction of the EEP-Compound–Hyperthyroidism-Oxidative Stress-Target Network

The prototypes of EEP obtained through UPLC-Q-TOF-MS analysis were identified for network pharmacology analysis. After target fishing, a total of 481 EEP-related targets were identified. “Hyperthyroidism” and “Oxidative Stress” were the keywords used to search for mechanism-related targets, and 1237 hyperlipidemia-related targets and 9629 oxidative stress-related genes were obtained from GeneCards and DisGeNET, respectively. Then, the 102 overlapping targets between EEP-related targets, hyperthyroidism-related targets, and oxidative stress-related genes were selected as potential targets for EEP antioxidative against hyperthyroidism using the Venn tool (Figure 8A).

The EEP-compound–hyperthyroidism-oxidative stress–target network was constructed to explore the relationships among the 18 active compounds and 102 overlapping targets by Cytoscape 3.8.2 software. As shown in Figure 8B, the EEP-compound–hyperthyroidism-oxidative stress–target network contains a total of 102 nodes (18 compound nodes, 1 disease node and 83 target nodes) and 607 edges. In this network, the yellow diamonds represent prototypes and the green ellipses represent potential targets. The number of connections of a node is defined as the degree (edges), which is proportional to the importance of the node. Alisol A (degree = 40) has the highest node value and was considered the main active compound in the treatment of hyperlipidemia, a finding consistent with a previous report.

#### 2.6.2. Functional Enrichment and Pathway Analysis of the Predicted Targets for EEP Antioxidative against Hyperthyroidism

In the GO functional enrichment analysis, the results showed that 35 BP terms, 10 CC terms, and 23 MF terms were significantly enriched. The results of the top 10 GO analysis in Figure 9A revealed that, in BP, targets were abundant in the positive regulation of cell proliferation, positive regulation of transcription from RNA polymerase II promoter, and protein phosphorylation. In CC, targets were abundant in the membrane, cytoplasm, and plasma membrane. In MF, targets were abundant in metal ion binding, ATP binding, and transferase activity. Furthermore, the results of the KEGG analysis showed that the predicted targets for EEP antioxidative against hyperthyroidism were significantly enriched in the HIF signaling pathway, the AMPK signaling pathway, and other pathways in Figure 9B.

### 2.7. Conjoint Analysis of the Antioxidant Effect of EEP on Hyperthyroidism-Induced Oxidative Stress by Transcriptomics and Network Pharmacology Analysis

To further explore the antioxidative mechanism of EEP caused by hyperthyroidism, the Gene Target-Prediction Target network was established by network pharmacology combined with transcriptomics analysis. A total of 41 key genes, 100 potential targets, and 2 common targets were imported into Cytoscape to define the Gene Target-Prediction Target network and are shown in Figure 10A. FASN and HMGCR (found using an integrated transcriptomics analysis and network pharmacology approach) may play essential roles in the antioxidative mechanism of EEP in hyperthyroidism. HMGCR was confirmed via RT-qPCR in Figure 7C.

Common pathways between network pharmacology and transcriptomics analysis were selected as potential mechanism pathways for EEP antioxidative effect against hyperthyroidism by the Venn tool in Figure 10B. Three pathways were selected, namely the PPAR signaling pathway, the AMPK signaling pathway, and alcoholic liver disease. Considering that alcoholic liver disease is too broad, the PPAR signaling pathway and the AMPK signaling pathway were chosen as the mechanisms of the antioxidant effect of EEP on hyperthyroidism-induced oxidative stress for further research.

### 2.8. EEP Activate Protective Antioxidant Mechanisms on Oxidative Stress Induced by Hyperthyroidism via Ampk and Ppar Signalling Pathways

As shown in Figure 11B, five key EEP targets (ACADM, EHHADH, FABP, HMGCR, and SLC27A2) in AMPK and PPAR signaling pathways as defined by the KEGG enrichment. FABP and HMGCR with changes in protein expression were confirmed via western blotting in Figure 11A. EHHADH, HMGCR, and SLC27A2 with changes in gene expression were confirmed via RT-qPCR in Figure 7C. This suggests that the mechanism of the antioxidant effect of EEP on hyperthyroidism-induced oxidative stress via AMPK and PPAR signaling pathways is shown in Figure 11B.

## 3. Discussion

Decreases in thyroid hormone levels such as TSH, as well as excessive secretion of free thyroid hormones such as T3 or T4 in the blood circulation from the thyroid gland, cause hyperthyroidism [37]. Hyperthyroidism is easily achieved in rodent models by continuous treatment with thyroxine, a mixture of T3 and T4. Studies have shown that hyperthyroidism induced by thyroxine administration causes increased oxidative damage [38] in liver tissue [17]. In this work, we established an experimental hyperthyroidism mouse model following treatment with thyroxine, the thyroid hormone regulating the ability of EEP, and the hepatic oxidative stress markers were investigated to determine the antioxidation of EEP against oxidative damage caused by hyperthyroidism.

Of the two thyroid hormones, T3 has the higher metabolic activity [39]. In this study, thyroxine-induced EEP-treated animals exhibited low T3 levels, indicating that T3 was inhibited by EEP. In addition, the reduced TSH levels induced by thyroxine were also able to be increased by EEP. It is indicated that EEP can ameliorate the abnormal status of thyroid hormones in hyperthyroid mice. Along with this, the loss in body weight in thyroxine-treated mice was also evident, which may be due to lipolysis and protein degradation in an experimentally induced hyperthyroid state. Interestingly, the administration of hyperthyroid rats with the test EEP attenuated all the above-mentioned changes near to control values, indicating the beneficial role of EEP on hyperthyroidism.

On the other hand, EEP has an antioxidant effect on oxidative stress in the liver caused by hyperthyroidism. When thyroid hormones increase the body’s metabolic system [15], the resulting increase in the production of reactive oxygen species leads to lipid peroxidation [16]. The level of MDA, a representative product of lipid peroxidation, reflects the severity of oxidative stress [40]. SOD and CAT acting as antioxidant enzymes can prevent the harmful effects of oxidative stress, and the levels represent the oxidative activities in the body [41]. Results indicated that EEP can increase the levels of the antioxidant enzymes SOD and CAT and inhibit the increase of MDA after administration. It shows that EEP can improve the activity of antioxidant enzymes and the ability of directly scavenging oxygen free radicals and can alleviate lipid oxidative damage through antioxidation.

Subsequently, we explored EEP regulated the potential key genes associated with hyperthyroidism-induced oxidative stress at the transcript level with experimental validation. First, the DEGs of experimentally induced hyperthyroid mice compared to EEP-administered mice were determined, and then the DEGs were compared with oxidative stress genes to obtain the genes for EEP action on hyperthyroidism-induced oxidative stress, called EEP-Genes. Then, these EEP-Genes were used to perform module analysis to identify hub genes as key-genes using the clustering algorithm. Next, we tested the expression of 10 key-genes and we were further validated by RT-qPCR, among which seven key-genes, namely EHHADH, ACOT2, ACOT8, SLC27A2, EPHX2, ACACA and HMGCR were identified.

Specifically, thyroid hormones accelerate the basal metabolic rate and oxidative metabolism by induction of specific mitochondrial enzymes and can affect the glucose and lipids balance and lipid metabolism, including synthesis, mobilization, and degradation, resulting in oxidative stress. Acryl-CoA thioesterases (ACOTs) have been suggested to regulate the metabolism of fatty acids [42]. Since ACOTs involve 3,5,3′-triiodo-L-thyronine-induced mechanisms of lipid absorption, storage, and utilization, they are expected to have an impact on the regulation of metabolic processes [43]. Enzymes known as ACOT are responsible for hydrolyzing fatty acyl-CoA to produce free fatty acids and CoA-SH [44]. As the oxidation capacity increases, ACOT2 changes the availability of an intracellular pool of AcylCoA (restricted by ACOT1) towards increased mitochondrial uptake of Acyl-CoA for β-oxidation and lipid synthesis [45]. Two important primary bile acid products (chenodeoxycholate and cholate) are produced due to the action of the ACOT8 gene, which is associated with bile acid synthesis and is related to fat emulsification in the diet [46]. The 27 member 2 SOF (SLC27A2) plays an essential role in lipid biosynthesis and fatty acid transport. According to studies, SLC27A2 alters the proto-oncogene c-Fos in differentiated thyroid cancer, which affects cell growth and differentiation [47]. Soluble epoxide hydrolase 2 (EPHX2) encodes SEH, a crucial gatekeeper enzyme, which influences the function of the lipid signal of various metabolites by breaking down epoxy fatty acids into compatible diols [48]. Lipid signal transduction is reduced by SEH activity [49]. EPHX2 inhibitors can promote the progression of melanoma and fibrosarcoma in mouse models by increasing the level of endogenous lipid mediators, which indicates that there is a relationship between EPHX2-lipid metabolisms [50]. Moreover, enoyl-CoA hydrostase and 3-hydroxyacyl CoA dehydrogenase (EHHADH) play an essential role in the metabolism of medium-chain β-oxidation of dicarboxylic acids, which regulate liver cholesterol biosynthesis [51], as well as acetyl-CoA carboxylase (ACACA) [52], which is the essential enzymes to produce fatty acids. At the same time, 3-Hydroxy-3-Methylglutaryl-CoA Reductase (HMGCR) affects lipid synthesis, and in the absence of HMGCR in cholesterol synthesis, adipocytes show reduced fat accumulation [53]. Hyperthyroidism is thought to accelerate the production of free radicals, leading to oxidative damage to lipids. In our work, EEP reduced hyperthyroidism-induced lipid peroxidation levels like MDA, enhanced intracellular scavenging enzymes such as SOD and CAT, and modulated the role of the above genes as antioxidants in experimentally induced hyperthyroidism. Thus, the antioxidant effect of key EEP-targeted genes (EHHADH, ACOT2, ACOT8, SLC27A2, EPHX2, ACACA, and HMGCR) could represent novel biomarkers for further research.

Furthermore, a network pharmacology-based system perspective was adopted to elucidate the mechanism of EEP on hyperthyroidism-induced oxidative stress at the molecular level. The network pharmacology-integrated transcriptomics analysis showed that FASN and HMGCR were key targets for the antioxidant effects of EEP. The results showed that HMGCR was significantly expressed at both the mRNA and protein expression levels. This means that HMGCR may be a key target for the mechanism of the antioxidant stress effect of EEP on hyperthyroidism, both at the transcriptome level and at the molecular level and warrants subsequent in-depth study. However, our RT-qPCR experiments failed to show any significant mRNA expression of FASN, despite its obvious trend.

The joint analysis showed that both AMPK and PPAR signaling pathways were present in the network pharmacology and transcriptomics KEGG enrichment results. The AMPK signaling pathway is essential for oxidative stress. The AMP-activated protein kinase (AMPK) plays an important role in lipid, energy, and carbohydrate metabolism in hepatocytes [54], and studies have shown that it can be interfered with by antioxidants and activated by ROS production [34]. In addition, peroxisome proliferator-activated receptors (PPARs) also play an important role in the regulation of oxidative stress processes [47]. Our results suggest that the mechanism of the antioxidant effect of EEP on oxidative stress induced by hyperthyroidism is related to the activation of the AMPK signaling pathway and the PPAR signaling pathway. Further analysis of the gene targets enriched by transcriptomics into these two signaling pathways and the key genes we screened out in the transcriptomics section of our analysis revealed that the ACADM, EHHADH, HMGCR, and SLC27A2 genes appeared in both signaling pathways, so we validated their protein expression and added the key target in the PPAR signaling pathway, called FABP, which regulates downstream fatty acid oxidative stress. Our results suggest that EEP attenuates hyperthyroidism-induced oxidative stress via the PPAR and AMPK pathways, with the main targets of action being by influencing the expression of FABP and HMGCR at the protein level and by affecting the expression of EHHADH, HMGCR, and SLC27A2 at the mRNA level.

This work showed an ameliorative effect of EEP on hyperthyroidism-induced oxidative stress, which may be related to the AMPK and PPAR pathways, but still has limitations. We only investigated the ameliorative effect of EEP on hyperthyroidism-induced oxidative stress in the liver, and more studies are needed in sites such as the heart or cerebral cortex, as both have been shown to induce oxidative stress in these tissues, which could be further studied in the future. Secondly, we have only carried out the identification of the in vitro components of EEP and not the in vivo components of EEP after administration to hyperthyroid mice, which we will carry out in future.

## 4. Materials and Methods

### 4.1. Experimental Design

The purpose of this study was to determine whether EEP exerts antioxidative effects of hyperthyroidism-induced oxidative stress and to explain the mechanism. The workflow is shown in Figure 12. First, the constituents of EEP were identified by UPLC-Q-TOF-MS analysis to ensure the research approach was reliable and convincing. Hyperthyroidism was experimentally induced in mouse models through exposure to thyroxine and the antioxidative activity of EEP was determined. Next, we explored the underlying mechanisms by mining key genes through transcriptomic studies and validating 10 of them by RT-qPCR and a network pharmacology based on the constituents of EEP identified by UPLC-Q-TOF-MS analysis was adopted. Finally, a combined transcriptomic and network pharmacology analysis was performed to identify the key signaling pathways of EEP on oxidative stress induced by hyperthyroidism, the PPAR signaling pathway and the AMPK signaling pathway, and the key targets in the pathways were investigated by RT-qPCR or Western Blotting to elucidate the mechanisms of action.

### 4.2. Preparation of Ethanolic Echinacea purpurea Extracts

As shown in Figure 13, the fresh aerial parts of *Echinacea purpurea* (L.) Moench was cut off and extracted six times with 70% ethanol at 80 °C three times, and incubated for 2 h, 2 h, and 1 h, respectively. The ethanol extracting solution was concentrated by reducing pressure and dried by vacuum. Finally, the ethanol dry powders of EP were obtained (Amway (Shanghai, China). Technology Development Co., Ltd. provided samples of ethanol dry powders of EP, Lot no. ZZJSHRD202106011). The dry powders of EP were dissolved in distilled water to obtain 50 mL a high concentration of 0.4 g/mL solution for alcoholic gavage of EP. To 25 mL of the high-dose EP liquid, 25 mL of distilled water was added to obtain a concentration of 0.2 g/mL solution for alcoholic gavage and was set as the medium EP dose. Finally, a low-dose alcoholic EP gavage was obtained by combining 25 mL of medium-dose EP gavage liquid and distilled water to a final volume of 50 mL to obtain a 0.1 g/mL solution.

### 4.3. UPLC-Q-TOF-MS Analysis

To identify the constituents of the EEP extracts UPLC-Q-TOF-MS was conducted using standard compounds [55]. The UPLC apparatus used was an Agilent 1290 (Agilent Technologies, Inc., Santa Clara, CA, USA) with Agilent Q-TOF 6545 LC/MS (Agilent Technologies, Inc., Santa Clara, CA, USA). A Waters CORTECS UPLC T3 (2.1 × 100 mm, 1.6 µm). The mobile phase ratio and flow rates were acetonitrile (A) and a 0.1% aqueous solution of formic acid (B). The gradient program was set as follows: 0–3 min, 5% A and 95% B; 3–38 min, 5–25% A and 95–75% B; 38–45 min, 25–40% A and 75–60% B; 45–70 min, 40–55% A and 60–45% B; 70–75 min, 55–70% A and 45–30% B; 75–90 min, 70–85% A and 30–15% B; 90–95 min, 85–95% A and 15–5% B; 95–98 min, 95% A and5% B; 98–98.1 min, 95–5% A and 5–95% B; 98.1–100 min, 5% A and 95% B. Mass spectrometry detection mode by ESI-negative/positive ion mode, mass range 50–1500, gas temp (°C) was 320, drying gas (L/min) was 8, nebulizer (psi) was 35, sheath gas temp (°C) was 350, sheath gas flow (L/min) was 11, Vcap (V) was 4000, nozzle voltage (V) was 1000, Fragmentor (V) was 175, and Skimmer (V) was 65.

### 4.4. Animal Experiments

Vital River Co., Ltd. (Beijing, China) provided Kunming mice, which were housed at a controlled temperature (22 ± 2 °C) and humidity (50% ± 10%) at a 12-h/12-h light/dark cycle. The mice were males and 6 weeks old, weighing between 18 and 22 g. All feasible measures were taken to reduce animal suffering.

#### 4.4.1. Induction of Hyperthyroidism

Mice were randomly divided into five groups (*n* = 8), including normal control (control), hyperthyroidism-induced control (Model), EEP extracts treatment groups (EEP group, 1, 2, or 4 g/kg, body weight [BW], namely EEP-L, EEP-M, and EEP-H, intragastrically (ig), 0.1 mL/10 g) in hyperthyroidism mice, and the propylthiouracil treatment group (PTU group, 5 mg/kg, ig) in hypothyroidism mice as a reference group. Hyperthyroidism was induced with thyroxine (160 mg/kg, ig) once daily for 14 days. Thyroxine was obtained from Shanghai Zhonghua Pharmaceutical Co. LTD. (Shanghai, China; batch No. 200602). PTU is a thioamide drug commonly used to treat hyperthyroidism and decreases the amount of thyroid hormone from the thyroid gland by inhibiting 5’-deiodinase, which converts T4 to the active form T3. Therefore, PTU was selected as a reference drug to treat hyperthyroidism.

#### 4.4.2. Body Weight and Organ Index

The weights of the thymus and spleen were also measured on the final day, and their relative weights were calculated as the percentage of body weight.

#### 4.4.3. Sample Collection

At the end of the study, all animals were sacrificed, and blood samples were collected and centrifuged (at 1200× *g* at 4 °C for 5 min). Serum was kept at −20 °C as aliquots for further biochemical assay of liver enzymes. The liver samples were collected immediately and immersed in −80 °C liquid nitrogen for homogenate preparation to be used in subsequent biochemical assays.

#### 4.4.4. Assessment of Thyroid Hormone in Hyperthyroidism Mice

Enzyme-linked immunosorbent assay (ELISA) kits [56] were used to measure thyroid stimulation hormone (TSH), triiodothyronine (T3), and thyroxine (T4) levels (Maiman, Shanghai, PR China). The kits were acquired from Nanjing Jiancheng Institute of Bioengineering, Inc. (Nanjing, Jiangsu, China).

#### 4.4.5. Determination of Hepatic Oxidative Stress Markers in Hyperthyroidism Mice

Take 200 mg of hepatic tissue homogenate. The concentration changes of SOD, CAT, and MDA in the liver tissue of mice in each group are measured by colorimetric method on the automatic biochemical instrument [33], which is operated according to the instructions of the kit. All kits were acquired from Nanjing Jiancheng Institute of Bioengineering, Inc. (Nanjing, Jiangsu, China).

### 4.5. Transcriptome Experiment and Analysis

We randomly selected three mice from each group as biological replicates and extracted their total RNA using the Trizol reagent kit. The mRNA was then enriched, and reverse-transcribed into cDNA after fragmentation. A second strand of cDNA was synthesized and then purified with a QiaQuick PCR extraction kit. Lastly, second-strand cDNA was combined with poly(A), ligated to Illumina sequencing adapters, and sequenced using Illumina HiSeq2500 by APExBIO Co. (Shanghai, China) [57]. After clean data were obtained, reads were mapped to the ribosome RNA (rRNA) database and then to the reference genome. The mapped reads of each sample were assembled, each transcription region was calculated, and its expressed abundance and variations were quantified. Calculated gene expression was directly used to analyze differences in gene expression among these groups. The DEGs between the hyperthyroidism model versus EEP extracts livers were respectively analyzed.

#### 4.5.1. Identifying the Antioxidant Effect of EEP on Hyperthyroidism-Induced Oxidative Stress Genes

We converted the RNA sequencing (RNA-seq) data into transcripts per kilobase million (TPM) expression profiles. Data analysis was performed using the limma package in R. The cut-off values for DEGs were set as |log FC| > 1, *p* < 0.05 to detect DEGs.

The common targets of the DEGs and the oxidative stress genes caused by hyperthyroidism were filtered using the Venn tool (https://bioinfogp.cnb.csic.es/tools/venny/index2.0.2.html, accessed on 22 June 2022). The intersection was considered the potential targets for the treatment of oxidative stress genes caused by hyperthyroidism with EEP at the transcriptome level.

#### 4.5.2. GO and KEGG Enrichment Analysis

The database used the org.Hs.eg.DB database file on the Bioconductor platform. To better understand the function of DEGs, we performed enrichment analysis using GO and KEGG tools. The GO pathway includes three parts: molecular functions (MF), cellular components (CC), and biological processes (BP). KEGG pathway enrichment analysis analyzes key pathways in which differential genes are mainly involved.

#### 4.5.3. Construction of a PPI Network and Analysis of Modules

To clarify the relationship between DEGs, DEGs were submitted to the STRING11.0 database (https://string-db.org/, accessed on 23 June 2022) to construct a PPI network. The species was set to Homo Sapiens with a threshold of ‘medium confidence’ (>0.400) was set for the hyperthyroidism group versus EEP to obtain additional genes, and the rest were set as default settings. The PPI relationship data were obtained and further visualized using Cytoscape software, version 3.9.1. The cytoHubba plugin was used to calculate the core subnet with Degree algorithms. Related induced genes were obtained by the cluster algorithm cytoHubba among PPI networks. Subsequently, we further validated the expression of the key 10 EEP-Genes in the hub network, in mice.

#### 4.5.4. Validation of Key Genes mRNA Expressions in the Mouse Liver

TRIzol reagent was used to extract Total RNA (Servicebio, Wuhan, China). Using a PrimeScript RT Reagent Kit, reverse transcription was performed (Servicebio, Wuhan, China). SYBR Premix Ex Taq (Servicebio, Wuhan, China) was used for RT-qPCR using a Real-Time PCR System (Bio-Rad, CFX, Hercules, California, CA, USA). Wuhan Servicebio Technology Co., Ltd. designed and synthesized all primers. The internal reference was GAPDH. Appendix A lists the primers used in the RT-qPCR study. The fold changes of the indicated genes were calculated using the 2−^ΔΔ^Ct method [58].

### 4.6. Network Pharmacology Analysis

#### 4.6.1. Target Prediction of the Active Compounds

The potential active compounds of EEP detected by UPLC-Q-TOF-MS analysis were used as chemical information for network pharmacology research [59]. All SMILES formats of compounds were retrieved by PubChem (https://pubchem.ncbi.nlm.nih.gov/, accessed on 22 June 2022). The predicted targets of active compounds were obtained from the following databases: the Pharm Mapper (https://www.lilab-ecust.cn/pharmmapper/, accessed on 23 June 2022) and the Swiss Target Prediction (https://www.swisstargetprediction.ch/, accessed on 23 June 2022). All predicted targets of active compounds were standardized into official gene symbols using the UniProt database (https://www.uniprot.org/, accessed on 24 June 2022).

#### 4.6.2. Collection of Oxidative Stress Genes Caused by Hyperthyroidism

The information of the disease genes was collected by searching for keywords “hyperthyroidism” and “oxidative stress” in GeneCards (https://www.genecards.org/, accessed on 25 June 2022) (using the correlation score > 1), DisGeNET (https://www.disgenet.org/, accessed on 25 June 2022) (using the correlation score > 0.1), and the therapeutic target database (TTD, https://db.idrblab.net/ttd/, accessed on 26 June 2022).

The common targets of the EEP and the oxidative stress genes caused by hyperthyroidism were filtered using the Venn tool (https://bioinfogp.cnb.csic.es/tools/venny/index2.0.2.html, accessed on 28 June 2022). The intersection was considered the potential target for the treatment of oxidative stress genes caused by hyperthyroidism with EEP at the molecular level.

#### 4.6.3. GO and KEGG Enrichment Analysis

To decipher the mechanism of the potential targets, gene ontology (GO) functional enrichment analysis and Kyoto Encyclopedia of Genes and Genomes (KEGG) pathway enrichment analysis was performed using DAVID (https://DAVID.ncifcrf.gov/, version 6.8, accessed on 3 July 2022). The DAVID database combined with FunRich software (https://www.funrich.org/, version 3.1.4, accessed on 3 July 2022) was used to analyze and visualize the results of GO including biological process (BP), molecular function (MF), and cellular component (CC) ontologies. Meanwhile, the KEGG pathway was analyzed and annotated.

### 4.7. Combined Analysis of Transcriptomics and Network Pharmacology

To further explore the antioxidative mechanism of EEP on hyperthyroidism, Cytoscape was used to perform an integrated key targets and the common KEGG pathway were filtered using the Venn tool between network pharmacology and transcriptomics analysis. Based on the pathway network, five related genes were considered key targets for further validation by western blotting.

### 4.8. Determination of Protein Target in AMRK Signal Pathway and PPAR Signal Pathway Using Western Blotting

The total proteins of the liver were extracted from a standard lysis buffer containing protein ase inhibitors, and then the protein concentration in the lysates was determined by a BCA protein assay kit (cat: G2026, Servicebio, Wuhan, China). After denaturing the proteins via the boiling method, equal amounts of proteins were separated by 10% sodium dodecyl sulfate–polyacrylamide gel electrophoresis (SDS–PAGE) and transferred onto a polyvinylidene difluoride (PVDF) membrane. The membranes were blocked with nonfat milk (5%) for 1 h at room temperature and incubated with anti-HMGCSR rabbit pAb (cat: ab137043, abcam, Cambridge, UK), anti-ACADM rabbit pAb (cat: 55210-1-AP, proteintech, Chicago, IL, USA), anti-EHHADH rabbit pAb (cat: 26570-1AP, proteintech, Chicago, USA), anti-FABP rabbit pAb(cat: ab279649, abcam, Cambridge, UK) and anti-SLC27A2 rabbit pAb (cat: 14048-1-AP, proteintech, Chicago, IL, USA) at 4 °C overnight. All the antibodies were purchased from proteintech and abcam [60].

### 4.9. Statistical Analysis

The results represent at least three separate investigations (or eight mice in each group). SPSS 21.0 was used to perform statistical analysis on the data, which were expressed as the mean plus standard error of the mean (SEM) (GraphPad, La Jolla, CA, USA). One-way analysis of variance (ANOVA) was used to compare multiple groups of data, followed by the least-significant difference (LSD) post-hoc test or Dunnett T3 test for comparison of multiple groups. Statistical significance was considered at *p* < 0.05. The correlation between the potential biomarkers and biochemical indices in the liver was carried out by the Person correlation analysis. Statistically significance was set at *p* < 0.05.

## 5. Conclusions

In summary, EEP successfully reversed the decrease in body weight, thyroid hormone imbalance, and oxidative stress in hyperthyroidism mouse models and proved the effectiveness of the antioxidative mechanisms of EEP. In addition, in EEP, UPLC-Q-TOF-MS analysis revealed the presence of several phenolic compounds, which studies have shown to have an anti-hyperthyroid effect and antioxidation. Therefore, we hypothesize that the amelioration of oxidative stress induced by hyperthyroidism by EEP is mainly due to the presence of these phenolic compounds. Furthermore, our study was the first to combine transcriptomic data with network pharmacology analysis to evaluate EEP activity. Key targeted genes and proteins in PPAR and AMPK signaling pathways in the hyperthyroidism-induced oxidative stress model were identified with experimental validation.

## Figures and Tables

**Figure 1 ijms-24-00187-f001:**
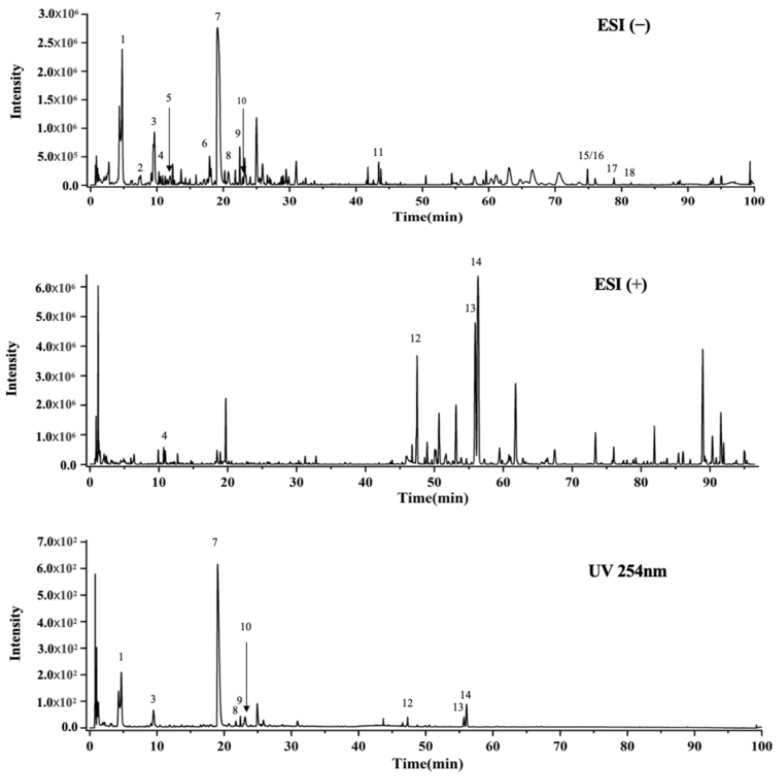
UPLC-Q-TOF-MS analysis of ethanolic extracts of *Echinacea Purpurea*. 1. Caftaric acid; 2. Coutaric acid; 3. Cis-fertaric acid; 4. Rhodioloside D; 5. Trans-fertaric acid; 6. N-Acetyl-DL-tryptophan; 7. Chicoric acid; 8. Quercetin 3-O-robinobioside; 9. Rutin; 10. Isoquercitrin; 11. Trihydroxyoctadecanoic acid; 12. 2,4-Undecadiene-8,10-diynoic acid isobutylamide; 13. Dodeca-2E,4E,8Z,10E-tetraenoic acid isobutylamide; 14. Dodeca-2E,4E,8Z,10Z-tetraenoic acid isobutylamide; 15. Hydroxyhexadecanoic acid; 16. Linolenic acid; 17. Linoleic acid; and 18. Palmitic acid.

**Figure 2 ijms-24-00187-f002:**
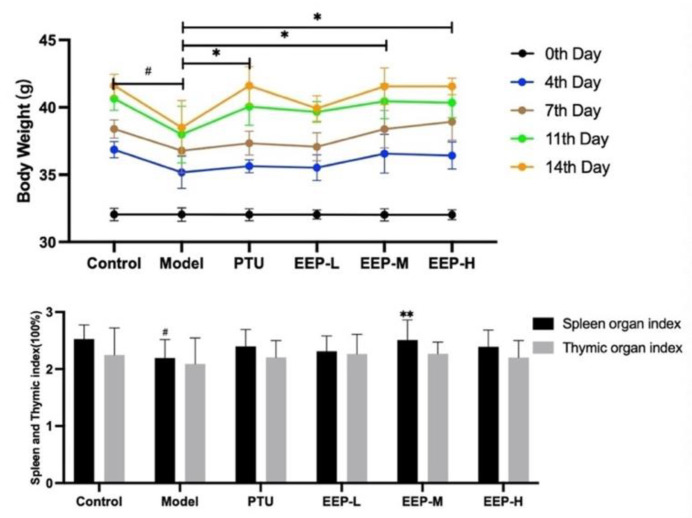
Effect of EEP treatment on changes in body weight, the thymus index, and the spleen index in hyperthyroidism mice. Data are presented as mean±SEM (*n* = 8). ^#^
*p* < 0.05versus Control. * *p* < 0.05, ** *p* < 0.01 versus Model.

**Figure 3 ijms-24-00187-f003:**
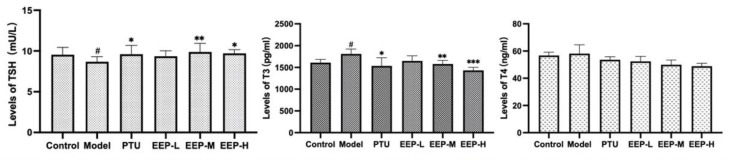
Effect of EEP treatment on thyroid hormone levels in mice with hyperthyroidism status. Data are presented as mean±SEM (*n* = 8). ^#^
*p* < 0.05versus Control; * *p* < 0.05, ** *p* < 0.01 and *** *p* < 0.001 versus Model.

**Figure 4 ijms-24-00187-f004:**
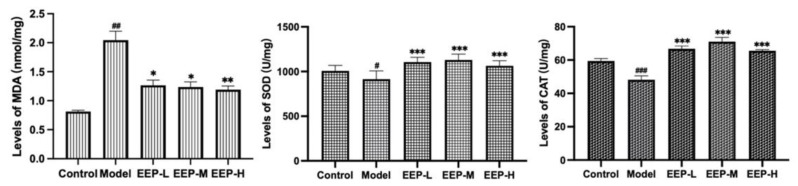
Effect of EEP treatment on antioxidative markers in hyperthyroidism mice. Data are presented as mean±SEM (*n* = 8). ^#^
*p* < 0.05, ^##^
*p* < 0.01 and ^###^
*p* < 0.001 versus Control; * *p* < 0.05, ** *p* < 0.01 and *** *p* < 0.001 versus Model.

**Figure 5 ijms-24-00187-f005:**
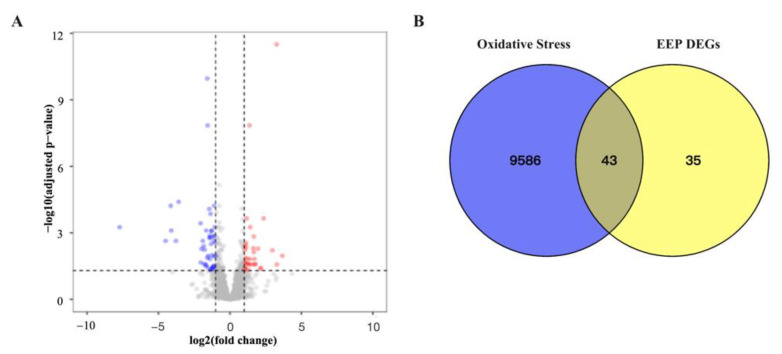
Identification antioxidative genes of differentially expressed genes of EEP in hyperthyroidism mice of EEP on transcriptomics profiles. (**A**) Volcano map of significantly upregulated genes and downregulated genes in the EEP and hyperthyroidism model groups. (**B**) Venn diagram of EEP against hyperthyroidism DEGs, oxidative stress, and intersection genes.

**Figure 6 ijms-24-00187-f006:**
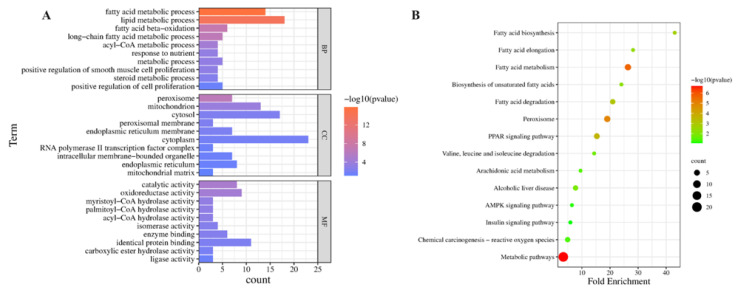
GO and KEGG enrichment analysis of potential genes for EEP antioxidative mechanism against hyperthyroidism of EEP on transcriptomics profiles. (**A**) Bar chart of GO analysis, (**B**) bubble chart of KEGG pathways.

**Figure 7 ijms-24-00187-f007:**
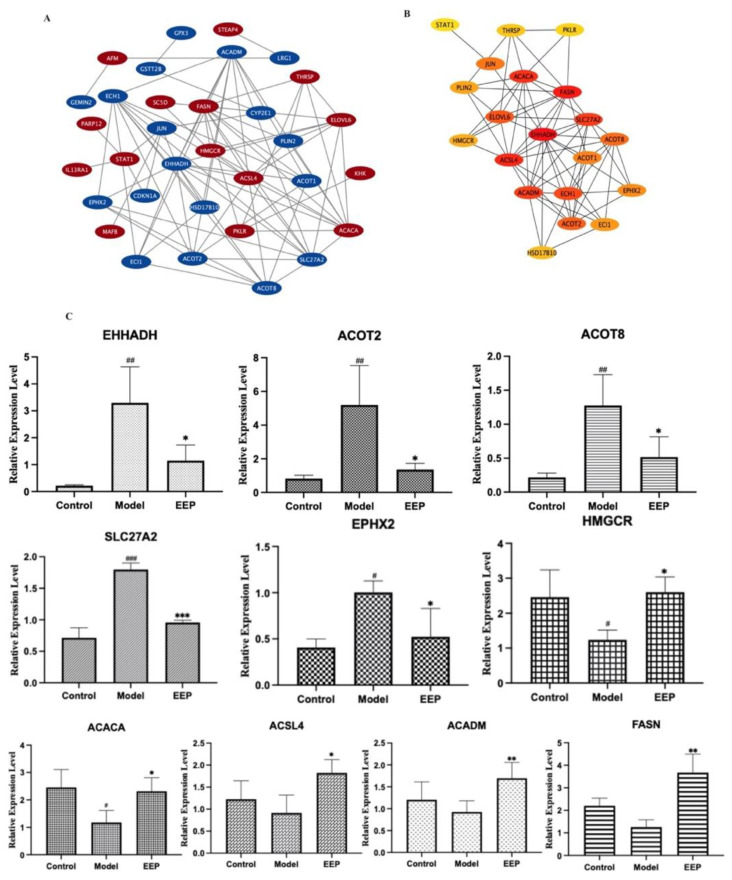
Significant modular analysis and validation based on the PPI network of antioxidative perturbation genes of EEP. (**A**) PPI network of EEP-induced genes in the controls compared with the hyperthyroidism model and treatment with EEP. Red represents the upregulated genes, and blue represents the downregulated genes. (**B**) Top 20 genes defined by cytoHubba plugin degree cluster analysis. The ellipses and the lines represent genes and the interaction of genes between genes, respectively, the color of the ellipse varies from yellow to orange to red, representing the degree of low to high interaction. (**C**) Related mRNA expression levels of the key 10 perturbation antioxidative genes of EEP for hyperthyroidism in hyperthyroidism status mice. Data are presented as mean±SEM (*n* = 3). ^#^
*p* < 0.05, ^##^
*p* < 0.01 and ^###^
*p* < 0.001 versus Control. * *p* < 0.05, ** *p* < 0.01 and *** *p* < 0.001 versus Model.

**Figure 8 ijms-24-00187-f008:**
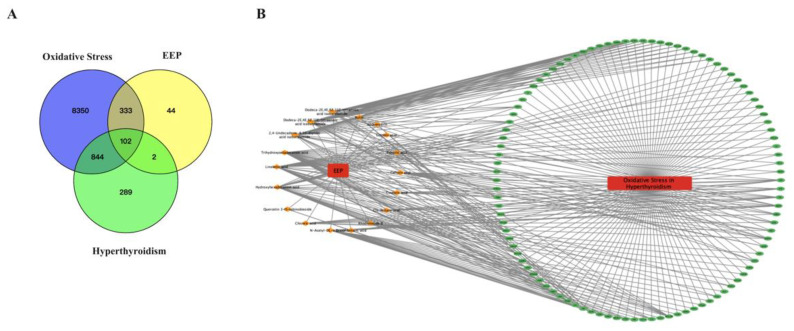
Influence of antioxidative mechanism of targets for EEP antioxidative against hyperthyroidism. (**A**) Venn diagram of EEP, oxidative stress and hyperthyroidism intersection targets; (**B**) EEP-compound–hyperthyroidism-oxidative stress–target network.

**Figure 9 ijms-24-00187-f009:**
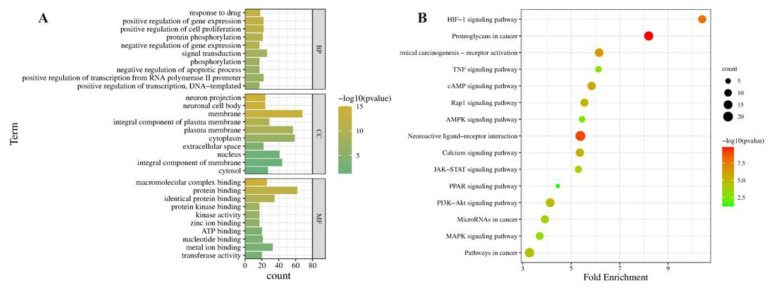
GO and KEGG enrichment analysis of potential targets for EEP antioxidative mechanism against hyperthyroidism of EEP. (**A**) bar chart of GO analysis, (**B**) bubble chart of KEGG pathways.

**Figure 10 ijms-24-00187-f010:**
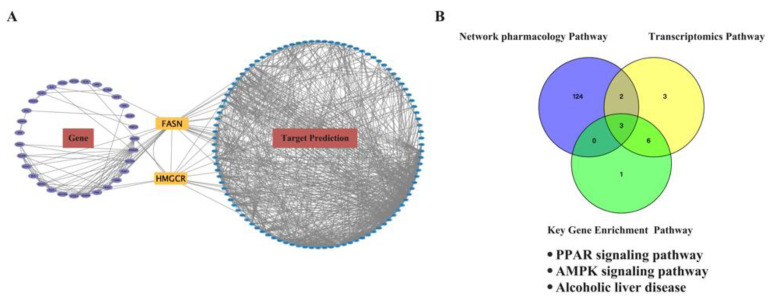
Influence of antioxidant effect of EEP on hyperthyroidism-induced oxidative stress on transcriptomics profiles and network pharmacology analysis. (**A**) Gene Target-Prediction Target network. The purple ellipses represent the key targets of antioxidant gene against hyperthyroidism from transcriptomics, the blue ellipses represent the key targets of antioxidant protein targets against hyperthyroidism from network pharmacology, and the orange rectangle represent common gene targets. (**B**) Common pathways of transcriptomics and network pharmacology analysis.

**Figure 11 ijms-24-00187-f011:**
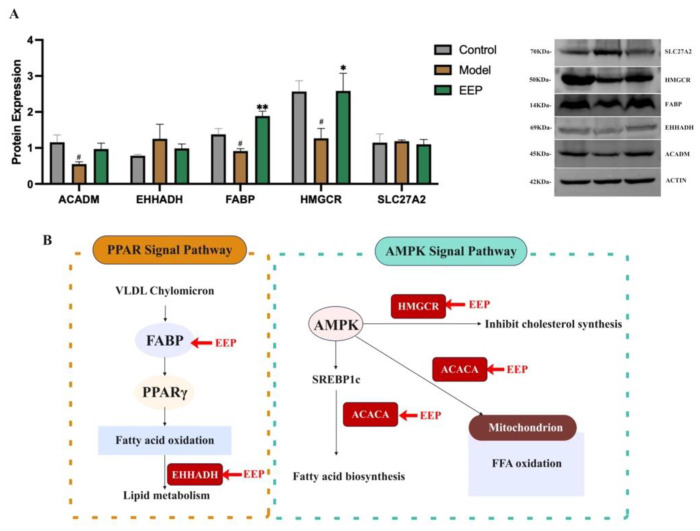
Antioxidant mechanisms of EEP via the AMPK and PPAR signaling pathways after hyperthyroidism challenge. (**A**) Effect of EEP on expressed of the AMPK and PPAR signaling pathways in hyperthyroidism mice (*n* = 3). The protein expressed of ACADM, EHHADH, FABP, HMGCR, and SLC27A2 in liver. ^#^
*p* < 0.05 compared with Control group; * *p* < 0.05, ** *p* < 0.01 compared with Model group. Data were expressed as means ± SEM. (**B**) EEP activate protective antioxidant mechanisms on oxidative stress induced by hyperthyroidism via the AMPK and PPAR signaling pathways.

**Figure 12 ijms-24-00187-f012:**
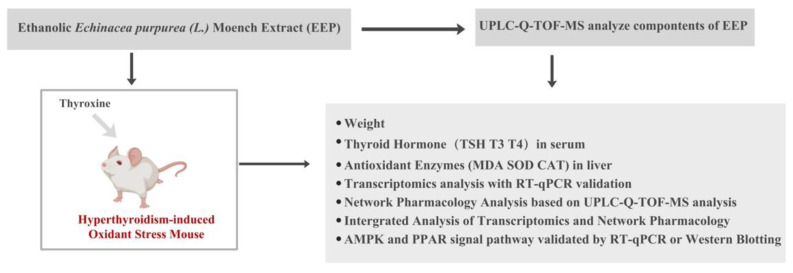
Workflow of methodologies applied in the study.

**Figure 13 ijms-24-00187-f013:**
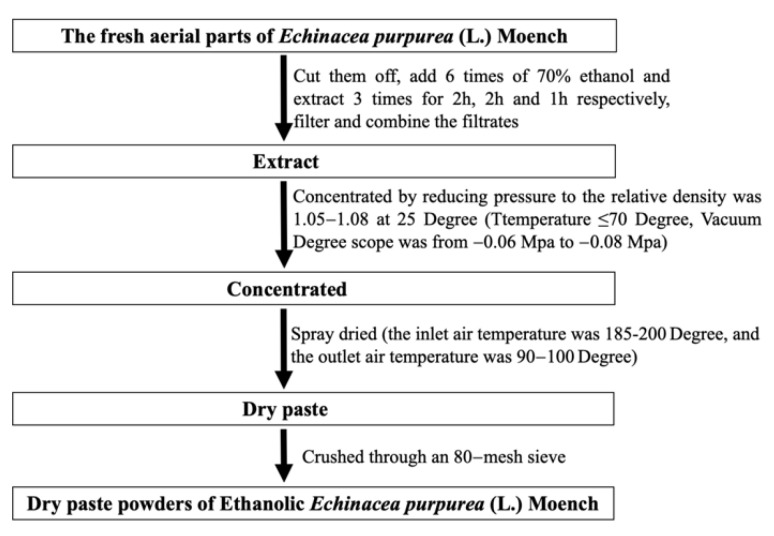
The flow diagram on the preparation of ethanolic *Echinacea purpurea* extracts.

## Data Availability

The original contributions presented in the study are included in the article/Appendix A, further inquiries can be directed to the corresponding authors.

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
