# Peer review of "Ameliorative Effect of Ethanolic Echinacea purpurea against Hyperthyroidism-Induced Oxidative Stress via AMRK and PPAR Signal Pathway Using Transcriptomics and Network Pharmacology Analysis"

_ijms, 2022, doi:10.3390/ijms24010187_

Round 1
Reviewer 1 Report
Good research has been conducted and presented by authors
Introduction may be condensed.
Conclusion may be extended
figures especially gel pictures may be incorporated in main text
Reviewer 2 Report
1- It is suggested to illustrate the mechanism PPAR signalling pathway and AMRK signalling pathway, that were considered the crucial signal pathways of the antioxidative mechanism of EEP against hyperthyroidism-induced oxidative stress. At the end of introduction you can add it and improve the manuscript.
2-It is suggest to indicate the bioactive compounds in Figure 1. UPLC-Q-TOF-MS analysis of ethanolic extracts of Echinacea Purpurea. This is usually the standard way so that the audience will understand easily and clearly.
3-Please check all raw file. Some file i cannot able to open it. Please check it before submit the manuscript. I hugely appreciate you include all raw data as supplementary file that stengthen the evidence.
4- Some pictures are blurred. for examples
Figure 7. Significant modular analysis and validation based on the PPI network of antioxidative perturbation genes of EEP. (A)(B)
5-Can you make an effort the strengthen the result with better discussion by inclusion of all key reference.
6- The botanical name must always be italic format. Check your manuscript.
7-Please draw the flow diagram on Preparation of Ethanolic Echinacea purpurea Extracts.
8-Please explain grinding meth0dology of plants.
9-Please explain drying methodology of plant.
10-Please explain sieve methodology of cut plant.
11- Please explain percentage of extract obtain and calculations.
12-Please include the heading of instrumentation and provide all instrument used with source.
13- It is mandatory to cite the methodology of have adopted and/or modified.
14. The study involve the animal study so mandatory to adhere the ARRIVE guidelines. https://arriveguidelines.org/
15- Please download the ARRIVE CHECKLIST, Include it as supplementary file and add one statement that the manuscript adhare to ARRIVE GUIDELINES.
